# A CRISPR screen identifies genes controlling *Etv2* threshold expression in murine hemangiogenic fate commitment

Haiyong Zhao[1] & Kyunghee Choi [1,2]

The ETS transcription factor *Etv2* is necessary and sufficient for the generation of hematopoietic and endothelial cells. However, upstream regulators of *Etv2* in hemangiogenesis, generation of hematopoietic and endothelial cells, have not been clearly addressed. Here we track the developmental route of hemangiogenic progenitors from mouse embryonic stem cells, perform genome-wide CRISPR screening, and transcriptome analysis of en route cell populations by utilizing *Brachyury*, *Etv2*, or *Scl* reporter embryonic stem cell lines to further understand the mechanisms that control hemangiogenesis. We identify the forkhead transcription factor *Foxh1*, in part through *Eomes*, to be critical for the formation of FLK1[+] mesoderm, from which the hemangiogenic fate is specified. Importantly, hemangiogenic fate is specified not simply by the onset of *Etv2* expression, but by a threshold-dependent mechanism, in which VEGF-FLK1 signaling plays an instructive role by promoting *Etv2* threshold expression. These studies reveal comprehensive cellular and molecular pathways governing the hemangiogenic cell lineage development.

[1] Department of Pathology and Immunology, Washington University School of Medicine, St. Louis, MO 63110, USA. [2] Developmental, Regenerative, and Stem Cell Biology Program, Washington University School of Medicine, St. Louis, MO 63110, USA. Correspondence and requests for materials should be addressed to K.C. (email: kchoi@wustl.edu)

Integration of the extrinsic signals into lineage-specific gene expression forms the basis for cell fate decisions. Accordingly, it is crucial to generate a comprehensive lineage map, to identify extrinsic cues that guide a specific cell lineage outcome and to delineate downstream signal cascades and transcriptional networks involved in lineage specification. Such information in turn would facilitate efforts deriving a desired cell type from pluripotent stem cells for regenerative medicine. To this end, hematopoiesis, the generation of blood, offers a unique model to study cell fate determination. While the lineage map downstream of the hematopoietic stem cells (HSCs) has been extensively described[1], it is still largely unknown how HSCs themselves are generated during embryogenesis.

Currently, it is well accepted that hematopoietic cells develop from mesoderm through hemangiogenic progenitors[2–4] and hemogenic endothelium intermediates[5–7]. The close developmental association between hematopoietic and endothelial cells is manifested by many transcription factors and signaling pathways that are commonly shared between these two cell populations. Gene-targeting studies have also shown that mutations in any of the shared genes often affect both cell lineages, supporting the notion of the common genetic pathway regulating hematopoietic and endothelial cell lineage development and function. Of these, Etv2 (aka Er71 and etsrp) has emerged as an obligatory factor, whose function is required at the earliest stage in hematopoietic and vascular development. In particular, Etv2 deficiency leads to embryonic lethality due to a complete block in blood and endothelial cell formation. Conversely, enforced Etv2 expression can ectopically activate both cell lineages[8–10]. These studies support the notion that Etv2 functions at the core of the common genetic pathway in blood and endothelial cell generation. Therefore, Etv2-controlled hemangiogenesis provides a powerful paradigm for modeling and assessing how exactly cell fate determination can be achieved by a single factor.

In vitro differentiation model of embryonic stem (ES) cells, which overcomes the cell number limitations for early embryonic studies and allows large-scale genetic screening, has been extensively used for studies pertaining to lineage development. To understand cellular pathways and molecular mechanisms regulating hemangiogenic cell lineage specification, here we utilized T/Brachyury, Etv2, and Scl expression together with PDGFRα and FLK1⁺ mesodermal markers to track hemangiogenic cell lineage development during ES cell differentiation. We performed transcriptome analysis of the transitional cell populations and high-throughput clustered regularly interspaced short palindromic repeats (CRISPR) screening[11] to further understand upstream molecular events of hemangiogenesis. Our data demonstrate a well-defined developmental route of hemangiogenesis, in which the forkhead transcription factor Foxh1 regulates, functioning in part through Eomes, FLK1⁺ mesoderm formation. The hemangiogenic fate is specified within FLK1⁺ mesoderm by the Etv2 threshold expression, which requires the VEGF-FLK1 signaling.

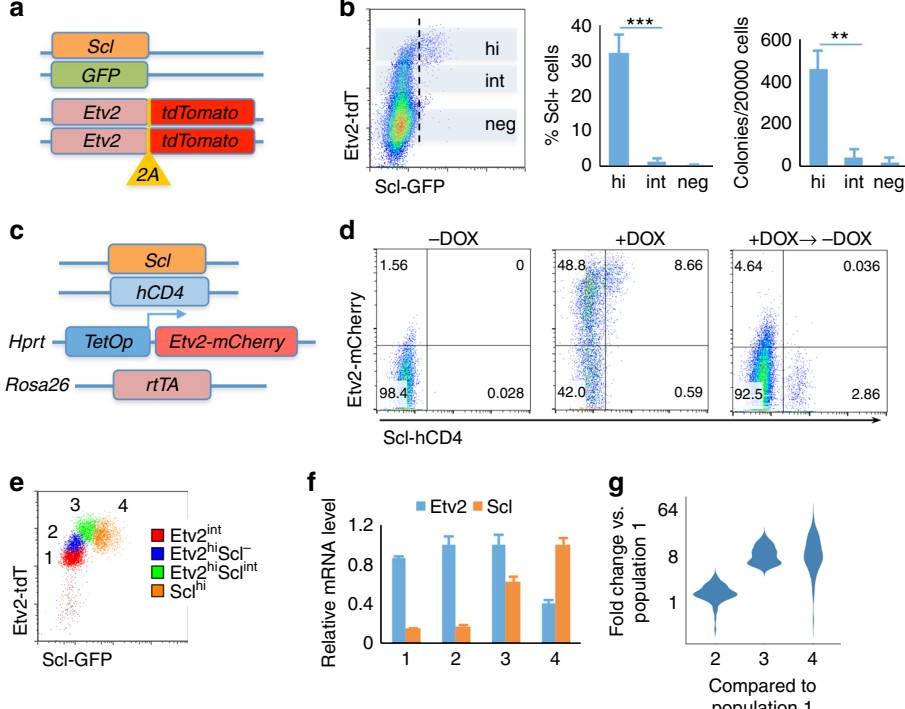

**Fig. 1** Etv2 threshold expression determines hemangiogenic fate. **a** Scheme of SGET ES cells. **b** Etv2-tdTomato and Scl-GFP expression in D4 SGET EBs analyzed by flow cytometry is shown on the *left*. The percentages of Scl-GFP⁺ cells (*right side* of the *vertical dashed line*) within Etv2-tdTomato^hi(gh), Etv2-tdTomato^int(ermediate) and Etv2-tdTomato^neg(ative) cells (*shaded regions*) are shown in the *middle*. Blast colony number obtained from sorted Etv2^neg, Etv2^int, and Etv2^hi cells is shown on the *right*. **c** Scheme of iEtv2-mCherry ES cell line. **d** iEtv2-mCherry EB cells differentiated in serum-free conditions were analyzed for Etv2-mCherry and Scl-hCD4. –DOX, no DOX control, +DOX, DOX was added from D2.5 to D3.5 at a concentration of 2 µg/mL, +DOX → –DOX, DOX was added from D2.5 to D3.5, washed off and then cells were cultured for additional 2 days. –DOX and +DOX EBs were analyzed on D3.5. +DOX → –DOX EBs were analyzed on D5.5. **e** Flow cytometry plot of the four populations, *1* Etv2-tdTomato^int, *2* Etv2-tdTomato^hi/Scl-GFP^-(negative), *3* Etv2-tdTomato^hiScl-GFP^int, and *4* Scl-GFP^hi from D4 SGET cells after sorting is shown. **f** Normalized relative mRNA level of Etv2 and Scl in the sorted populations is shown. The mRNA level of Etv2 and Scl was first normalized to Gapdh, and then the maximal value within the group was rescaled to 1. **g** Violin plots showing fold changes of RNA-seq RPKM values of 73 ETV2 target genes in populations 2, 3, and 4, respectively, compared to population 1. **P value <0.01 in Student's t test, ***P < 0.001, and n = 3. Error bars are s.d

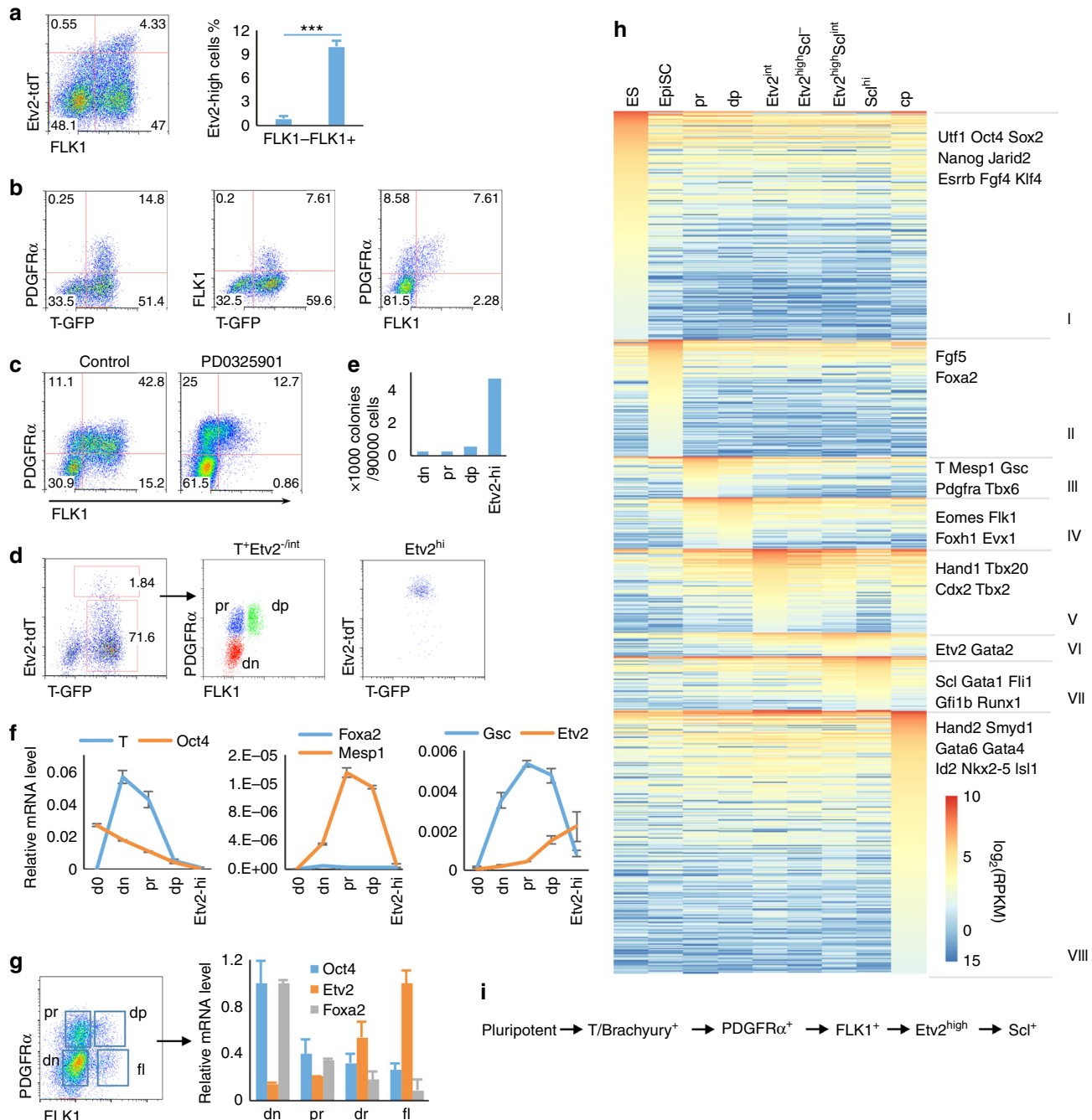

**Fig. 2** Developmental route of Etv2[hi] hemangiogenic progenitors. **a** FACS plot of Etv2-tdTomato and FLK1 in D4 SGET EBs is shown on the *left*. The percentage of Etv2[high] cells within FLK1-negative or -positive cells is shown on the *right*. ***P value <0.001 in Student's *t* test, n = 3. *Error bars* are s.d. **b** Coexpression patterns of PDGFRα vs. T-GFP, FLK1 vs. T-GFP, and PDGFRα vs. FLK1, in D2.5 TGET EBs. **c** D3.5 SGET EBs treated with DMSO (control) or ERK inhibitor PD0325901 (2 μM) were analyzed for PDGFRα and FLK1 expression by flow cytometry. PD0325901 was added from D2.5 to 3.5. **d** FACS sorting scheme of various mesodermal cell populations from D3.5 TGET EB is shown. T-GFP[+]PDGFRα[-]FLK1[-] (*dn*), T-GFP[+]PDGFRα[+]FLK1[-] (*pr*), and PDGFRα[+]FLK1[+] (*dp*). **e** Blast colony number obtained from various cell populations as sorted in **d** is shown. **f** Marker gene expression pattern in the sorted populations as in **d** is shown. D0 indicates undifferentiated ES cells. **g** Indicated populations were sorted from E7.5 embryos and analyzed for relative mRNA levels. **h** Comparative transcriptome analysis for indicated cell populations is shown. Three thousand eight hundred fifteen genes that are expressed in at least one sample and have a minimal RPKM change of 5-fold among the samples were chosen for comparison. *ES* undifferentiated mouse embryonic stem cells, *EpiSC* ES cell-derived epiblast stem cells, "pr" and "dp" were already described in **d**; Etv2[int], Etv2[hi]Scl[-], Etv2[hi]Scl[int], and Scl[hi] cells were described in Fig. 1e; "cp", ES cell-derived cardiac progenitors[25]. The D6 endoderm sample is not shown in the plot, but is included in the differential expression analysis and Supplementary Data 2. **i** Schematic diagram shows a developmental route of ES cells to the Etv2-tdTomato[hi(gh)]/Scl-GFP[+] hemangiogenic state

## Results

### Etv2 threshold expression determines hemangiogenic fate.

Given that *Etv2* functions at the core of the genetic pathway in the generation of hemangiogenic progenitor cells[8-10], we reasoned that tracking its expression would help delineate molecular and cellular events leading to hemangiogenic cell lineage specification. Thus, we established a reporter ES cell line expressing tdTomato and GFP from the *Etv2* and *Scl* loci, respectively, to monitor endogenous *Etv2* and *Scl* expression (SGET ES cells, Fig. 1a). *Scl* is a direct ETV2 target[10, 12, 13] and is essential for hematopoietic lineage development[14]. As expected, the onset of Scl-GFP expression in differentiating ES cells (embryoid bodies, EBs) was later than that of Etv2-tdTomato (Supplementary Fig. 1a). Importantly, emerging Scl-GFP$^+$ cells were mainly observed within cells expressing high levels of *Etv2* (Etv2-tdTomato$^{high}$), suggesting an ETV2 threshold requirements in target gene expression (Fig. 1b, *left* and *middle*). Consistently, blast-colony-forming cells (BL-CFCs, representing hemangiogenic potential in vitro)[3, 15] were enriched in Etv2-tdTomato$^{high}$ cells, confirming that hemangiogenic fate is specified in Etv2-tdTomato$^{high}$ cells (Fig. 1b, *right*). *Etv2*-heterozygous cells still followed the threshold rule (Supplementary Fig. 1b). The Scl-GFP$^+$ cells formed obvious bimodal distribution with time (Supplementary Fig. 1c), suggesting an autoregulatory feedback mechanism for *Scl* expression[16]. The hematopoietic marker CD41 and the endothelial cell marker TIE2 expression were observed within Scl-GFP$^+$ cells (Supplementary Fig. 1c).

To further validate the threshold requirement of *Etv2* expression for the initiation of *Scl* expression, we established an ES cell line that expresses doxycycline (DOX)-inducible *Etv2-mCherry*, and human CD4 surface antigen from the endogenous *Scl* locus[17] (iEtv2-mCherry ES cells, Fig. 1c). Enforced *Etv2-mCherry* expression activated Scl-hCD4 and TIE2 in serum-free conditions, a differentiation system which normally does not generate mesoderm cells unless BMP4 is supplemented[17]. Importantly, an obvious threshold requirement of *Etv2* could be reconstituted in this condition (Fig. 1d, *left* and *middle*, and Supplementary Fig. 1d). *Etv2* expression level in Etv2-mCherry$^{high}$Scl-hCD4$^-$ cells were higher than in Etv2-tdTomato$^{high}$Scl-GFP$^-$ cells (Supplementary Fig. 1e). This suggests that the exact threshold level of *Etv2* expression for activating *Scl* in different systems (endogenous vs. exogenous or serum vs. serum free) may be different. Notably, Scl-hCD4$^+$ cells were sustained after DOX withdrawal (Fig. 1d, *right*), indicating establishment of an autoregulatory mechanism of *Scl* expression.

To confirm that hemangiognei program is initiated only after the *Etv2* threshold expression is achieved, we sorted four populations from SGET EBs, Etv2-tdTomato$^{int(ermediate)}$, Etv2-tdTomato$^{hi(gh)}$Scl-GFP$^-$, Etv2-tdTomato$^{hi(gh)}$Scl-GFP$^{int(ermediate)}$, and Scl-GFP$^{hi(gh)}$ (Fig. 1e). RT-qPCR showed that mRNA levels of *Etv2* were consistent to tdTomato reporter signals, while *Scl* was activated mainly after *Etv2* threshold was fulfilled (Fig. 1e, f, *left*). We subsequently performed RNA-seq analysis of the four populations. Out of genes that were upregulated at least fivefolds in Etv2-tdTomato$^{high}$Scl-GFP$^{int}$ cells compared to Etv2-tdTomato$^{int}$ cells, 73 have been previously identified to be ETV2 targets that included almost all the important hemangiogenic genes[10]. The expression of these 73 ETV2 target genes became fully activated predominantly in Etv2-tdTomato$^{high}$Scl-GFP$^+$ (Scl-GFP$^{int}$ or Scl-GFP$^{high}$) cells (Fig. 1f, *right*, and Supplementary Data 1), supporting the notion that the hemangiogenic program becomes activated after the *Etv2* threshold expression is achieved. Collectively, *Etv2* directly specifies the hemangiogenic fate in a threshold-dependent manner.

### Developmental route to Etv2$^{high}$ hemangiogenic progenitors.

We next determined the developmental route generating Etv2-tdTomato$^{high}$ hemangiogenic progenitor cells from pluripotent cells. As *T/Brachyury* expression is first detected in the primitive streak/mesendoderm and persists in early mesoderm and endoderm[18], we established another ES cell line expressing GFP and tdTomato from the endogenous *T* and *Etv2* loci, respectively (TGET ES cells, Supplementary Fig. 2a). To track *Etv2* expression dynamics during differentiation, we utilized two additional mesoderm markers, the PDGF receptor α[19, 20] and the VEGF receptor FLK1[21, 22]. While Etv2-tdTomato$^{int}$ cells were observed in both FLK1$^-$ and FLK1$^+$ cells, Etv2-tdTomato$^{high}$ cells were found predominantly within FLK1$^+$ cells (Fig. 2a). Both FLK1$^+$ and PDGFRα$^+$ cells were found within T-GFP$^+$ cells in early EBs, day 2.75, confirming their primitive streak origin (Fig. 2b, *left* and *middle*). At this stage, FLK1$^+$ cells were simultaneously PDGFRα positive, (Fig. 2b, *right*). Moreover, PDGFRα single positive cells could generate FLK1$^+$ cells and CD41$^+$ hematopoietic cells when sorted and cultured on OP9 stromal cells (Supplementary Fig. 2b). Furthermore, when differentiating EBs were treated with the Erk map kinase inhibitor PD0325901 on D2.5, there was a block in FLK1$^+$ cell generation from PDGFRα single positive cells (Fig. 2c). These results suggested that PDGFRα expression is a prerequisite for that of FLK1, despite the fact that PDGFRα$^-$FLK1$^-$, PDGFRα$^+$FLK1$^-$, and PDGFRα$^+$FLK1$^+$ cells were shown to be highly plastic and interchangeable[20].

To better delineate the relationship between *T*, *Pdgfra*, *Flk1*, and *Etv2*, we sorted TGET EB cells into T-GFP$^+$PDGFRα$^-$FLK1$^-$ ("dn"), T-GFP$^+$PDGFRα$^+$FLK1$^-$ (PDGFRα single positive, "pr"), T-GFP$^+$PDGFRα$^+$FLK1$^+$ ("dp"), and Etv2-tdTomato$^{high}$ cells ("Etv2$^{hi}$") (Fig. 2d). As expected, BL-CFCs were enriched in Etv2-tdTomato$^{high}$ cells (Fig. 2e). Analysis of marker gene expression by RT-qPCR indicated that the pluripotency gene *Oct4* and mesendodermal/endoderm marker gene *Foxa2* expression in "pr" cells was higher than in "dp" or "Etv2$^{hi}$" cells. *Etv2* expression was low in "dn" and "pr" cells, elevated in "dp" cells, and achieving its highest expression in "Etv2$^{hi}$" cells (Fig. 2f). Therefore, compared to "dp" cells, "pr" cells are more similar to "dn" cells, which might still represent the less differentiated mesendoderm stage. Analysis of marker gene expression in "dn", "pr", "dp", and "fl" (FLK1 single positive cells, which are enriched for hemangiogenic lineage cells[23]) populations from E7.5 embryos further supported the interpretation that "pr" cells are less differentiated than "dp" cells (Fig. 2g, *right*).

We next performed RNA-seq analysis of "pr" and "dp" cells, and compared the data to those published RNA-seq of undifferentiated ES cells, ES cell-derived epiblast stem cells (EpiSCs), endoderm and cardiac progenitors (Fig. 2h)[24-26]. We additionally included RNA-seq data from Etv2-tdTomato$^{int}$, Etv2-tdTomato$^{high}$Scl-GFP$^-$, Etv2-tdTomato$^{high}$Scl-GFP$^{int}$, and Scl-GFP$^{high}$ cell populations (described in Fig. 1e). Three thousand eight hundred fifteen genes were differentially expressed in the 10 samples (Fig. 2h and Supplementary Data 2). ES cells highly expressed pluripotency-related genes like *Nanog*, *Oct4*, and *Sox2* (group I), and EpiSCs expressed high levels of mesendoderm/gastrulation-related genes like *Foxa2* and *Fgf5* (group II). While genes expressed in the primitive streak/early mesoderm, such as *T*, *Mesp1*, *Gsc*, *Pdgfra*, *Tbx6*, and *Mesp2*, were enriched in group III ("pr" cells), genes enriched in groups VI and VII (hemangiogenic progenitors) or group VIII (cardiac progenitors) still retained low expression in "pr" cells. Therefore, gene expression profiling further suggests that "pr" cells represent nascent/early mesoderm. Consistently, "pr" transcriptome was more similar to the proximal regions of the primitive streak of the E7.0 gastrulating embryo (Supplementary Fig. 2c), where nascent mesoderm emerges, than "dp" cells, while

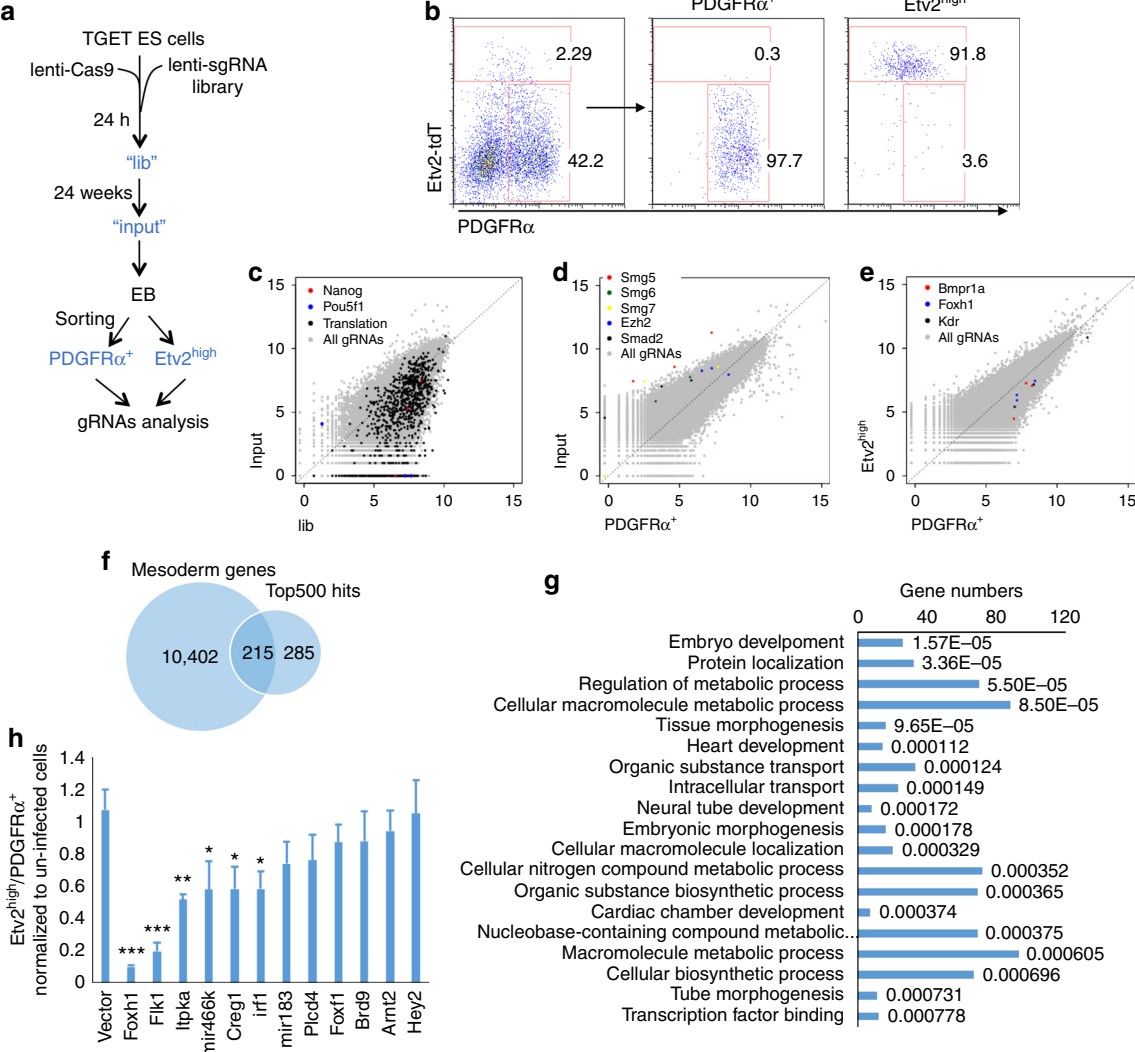

**Fig. 3** CRISPR screening identifies key *Etv2* upstream regulators. **a** Scheme for CRISPR screening. **b** FACS plot of the Etv2^high and PDGFRα^+ cells from D3.5 TGET EB cells used for cell sorting after CRISPR library infection is shown. **c** Comparative reads analysis between TGET ES cells (24 h post infection, "lib") and stabilized TGET ES cells (2 weeks selection post infection, "input"), which shows depletion of gRNAs against essential genes for pluripotency in stabilized TGET ES cells. **d** Comparative reads analysis between stabilized TGET ES cells ("input") and PDGFRα^+ cells, which shows depletion of gRNAs against essential genes for PDGFRα^+ mesoderm differentiation. **e** Comparative reads analysis between PDGFRα^+ and Etv2^high cells, which shows depletion of gRNAs against genes essential for Etv2^high cell generation. **f** Scheme for generation of the candidate gene list. Top 500 genes with depleted reads from PDGFRα^+ to Etv2^high cells were compared to 10,402 genes expressed in mesoderm (with RPKM >1 in any of "pr", "dp", Etv2^int, or Etv2^highScl^−), which yielded 215 genes. **g** Functional annotation of selected candidate genes is shown. Numbers at the end of each bar shows the *P* value of gene enrichment in the corresponding ontology item. **h** Validation of chosen candidate genes. The ratio of PDGFRα^+ to Etv2^high cells by a candidate sgRNA depletion was normalized to that of empty vector. Refer to Supplementary Fig. 3b, c for more details. *P* value <0.05 in Student's *t* test, **P < 0.01, ***P < 0.001, and n = 3. Error bars are s.d

transcriptome of ES cell-derived endoderm was more similar to the distal regions, which form definitive endoderm[27]. To further validate our transcriptome analysis, we additionally compared our data with a published transcriptome profiling of multiple stages in ES cell differentiation, which included ES cells, T^+FLK1^− cells ("mes"), FLK1^+ cells ("hb"), CD41^−Tie2^+Kit^+ hemogenic endothelial cells ("he"), and CD41^+ hematopoietic progenitor cells ("hp")[28]. The "mes" cells should contain both "dn" and "pr" populations, while the "hb" population should include all of the Etv2^−/int/hi cells in FLK1^+ populations. Data analysis revealed that "pr"-enriched genes (group III) were also enriched in "mes" cells in this published data set, with 36 out of 80 achieving their highest expression at this stage (Supplementary Fig. 2d, *left*, and Supplementary Data 3). Similarly, the 97 "dp"-enriched genes

(group IV) were also extensively expressed in "hb" cells, with 48 of them achieving their highest expression there (Supplementary Fig. 2d, *right*, and Supplementary Data 4). Strikingly, when only transcription-related genes were included[29], "mes" transcriptome was closer to "EpiSCs" than "pr", while the "hb" transcriptome fitted in right between "dp" and "Etv2^int" samples (Supplementary Fig. 2e and Supplementary Data 5). Meanwhile, the "hp" and "he" transcriptome were closer to our Etv2-tdTomato^highScl-GFP^+ sample (Supplementary Fig. 2e and Supplementary Data 5). From these analyses, we propose that hemangiogenic progenitors (ETV2^highSCL^+) develop from the pluripotent stage through the following developmental route: gastrulating mesendoderm (T-GFP^+), nascent/early mesoderm (PDGFRα^+), and lateral plate mesoderm (FLK1^+) (Fig. 2i).

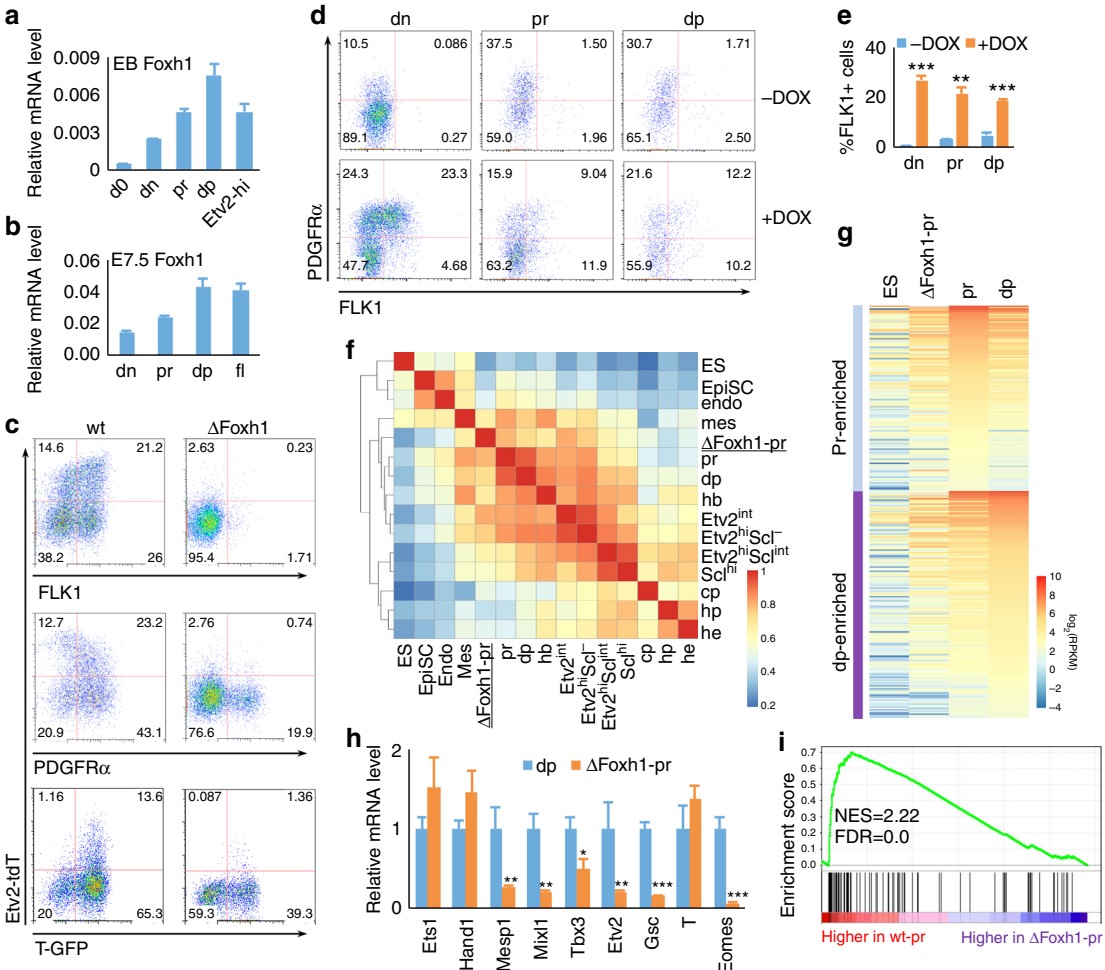

**Fig. 4** Foxh1 is required for FLK1⁺ mesoderm generation. **a** Relative mRNA levels of *Foxh1* in indicated cell populations sorted from D4 EBs are shown. **b** Relative mRNA levels of *Foxh1* in indicated cell populations sorted from E7.5 embryos are shown. **c** FACS analysis of D4 wild type (*wt*) and *ΔFoxh1* EB cells is shown. (*Top* and *middle*, SGET cells; *bottom*, TGET cells). **d** *iFoxh1-ΔFoxh1* cells were differentiated with 500 ng/mL DOX, "dn", "pr" and "dp" cells were sorted on D3, and re-seeded onto OP9 cells with or without 500 ng/mL of DOX. Cells were analyzed 24 h later by flow cytometry. The results are summarized in **e**. **f** Spearman correlation map of RNA-seq data of transcription-related factors from indicated populations. **g** Comparison of RNA-seq results of "pr-" or "dp"-enriched genes in *ΔFoxh1* "pr", wt "pr", "dp", and undifferentiated ES cells is shown. **h** Expression of key transcription factors with highest expression in "dp" (identified in Fig. 2g) was validated by qRT-PCR using *ΔFoxh1* "pr" and wt "dp" cells. **i** GSEA analysis of T-related genes[41] for downregulation in *ΔFoxh1*-pr vs. wt-pr cells. Only genes related to *T* with correlation scores >0.2 were chosen for GSEA analysis. *NES* normalized enrichment score, *FDR* false discovery rate. *P value <0.05 in Student's *t* test, **P < 0.01, ***P < 0.001, and n = 3. *Error bars are s.d*

**CRISPR screening identifies *Etv2* upstream signals**. Having defined the developmental route giving rise to Etv2^high^ hemangiogenic progenitor cells, we employed high-throughput loss-of-function screening utilizing the CRISPR/Cas9 system to identify upstream factors regulating generation of this population. We first established TGET ES cells stably expressing the Cas9 nuclease. Subsequently, we infected the cells with the GeCKO-v2 lenti-CRISPR virus library A, which contained 66,405 gRNAs against 20,608 mouse protein-coding genes and 1145 miRNAs[11]. We collected four samples from the library-infected cells (Fig. 3a, b): "lib", 24 h post infection, "input", 2 weeks post infection, "pr" and "Etv2^high^" cells sorted from "input"-derived EBs. The sample "lib" reflected the library complexity. While gRNAs against genes essential for cell viability or pluripotency maintenance should have been depleted in "input", gRNAs against genes required for Etv2-tdTomato^high^ cell generation but not for early mesoderm commitment should have been depleted specifically in "Etv2^high^" cells. We recovered and sequenced gRNAs from the four samples. The resulting gRNA counts and the ranks of individual genes were analyzed using HiTSelect,

an analysis tool for pooled RNAi or CRISPR screening[30] (Supplementary Data 6).

As expected, the top 1000 genes that were depleted in their gRNAs from "lib" to "input" were enriched for those involved in essential cell activities, such as DNA replication, transcription, mRNA splicing, and translation (Supplementary Fig. 3a). These also included the critical pluripotency-related genes such as *Oct4* and *Nanog* (Fig. 3c and Supplementary Data 7). From "input" to "pr", top ranked genes with depleted gRNAs included the lysine methyltransferase *Ezh2* of the polycomb group complex PRC2, and components of the nonsense-mediated mRNA decay pathway genes including *Smg5*, *Smg6*, and *Smg7* (Fig. 3d and Supplementary Data 8), which are required for early embryo development and ES cell differentiation[31]. *Smad2*[32], one of the transcription factors mediating TGFβ signaling to induce early mesoderm and endoderm commitment, also ranked high in the list.

High-ranking genes that were depleted in their gRNAs in Etv2^high^ cells compared to "pr" cells included the bone morphogenetic protein receptor *Bmpr1a*, which is essential for

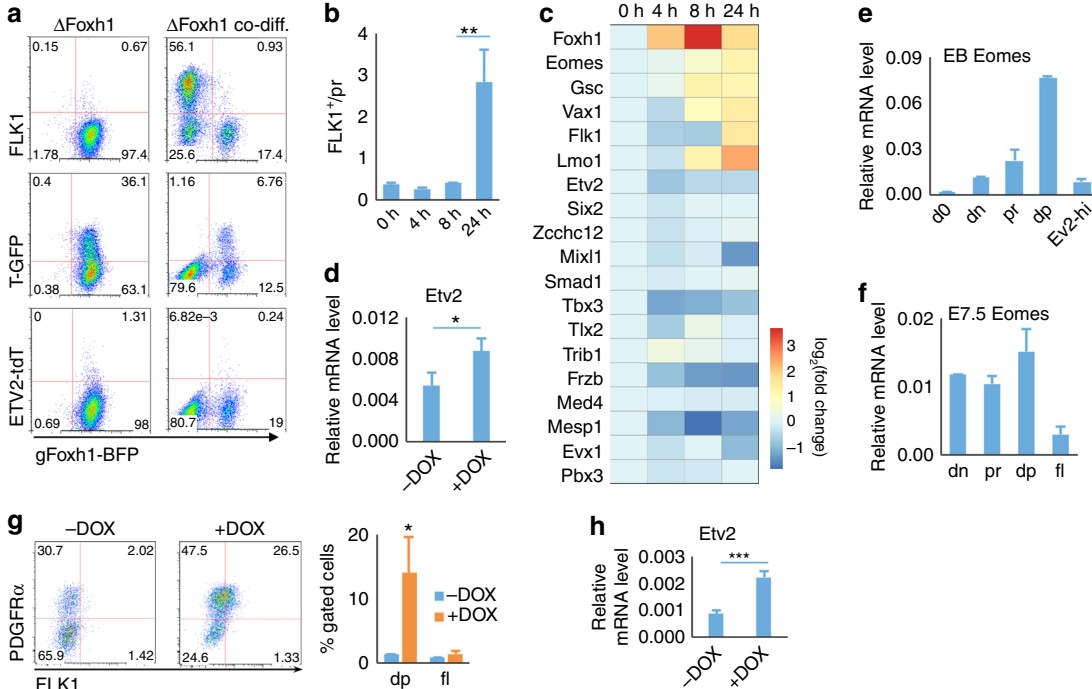

**Fig. 5** *Eomes* functions downstream of *Foxh1* in FLK1+ cell generation. **a** TGET *ΔFoxh1* ES cells (BFP positive) were differentiated either by themselves (*left*) or with wild-type A2lox ES cells (BFP negative, *right*). Subsequently, BFP-negative (wild type) or BFP-positive (*ΔFoxh1*) cells were analyzed for FLK1, T-GFP, and Etv2-tdTomato expression on D4. **b** iFoxh1-*ΔFoxh1* EB cells were treated with 500 ng/mL DOX for different time span as indicated and analyzed by flow cytometry on D3.5. **c** RT-qPCR analysis of indicated genes in D3.5 iFoxh1-*ΔFoxh1* EB cells treated with 500 ng/mL DOX for different time span. *Frzb* and *Smad1*, two signaling factors important for mesoderm development were included in the list. **d** RT-qPCR analysis of Etv2 was performed using iFoxh1-*ΔFoxh1* ES cells differentiated with or without DOX (500 ng/mL, D2.5–3.5), and then for additional 1 day without DOX. **e** *Eomes* mRNA levels in indicated populations sorted from D3.5 TGET EBs and undifferentiated ES cells ("d0") are shown. **f** *Eomes* mRNA levels are shown in indicated populations sorted from E7.5 embryos. **g** Flow cytometry analysis of D3.5 iEomes-*ΔFoxh1* EB cells treated with or without 50 ng/mL DOX for 24 h. **h** RT-qPCR analysis of Etv2 was performed using iEomes-*ΔFoxh1* ES cells differentiated with or without DOX (50 ng/mL, D2.5–3.5), and then for additional 1 day without DOX. *P value <0.05 in Student's *t* test, **P < 0.01, ***P < 0.001, and n = 3. Error bars are s.d

the generation of FLK1 and Scl-positive cells[33, 34], the VEGF receptor *Flk1* (Kdr), which is critical for hematopoietic and endothelial cell development[35], and the transcription factor *Foxh1*, which has been reported to be a cofactor for Smad2/3[36, 37] (Fig. 3e and Supplementary Data 9). By selecting genes highly expressed in mesoderm ("dn", "pr", or "dp" cells), we narrowed down 500 hits to 215 genes, which were enriched for development-related genes (Supplementary Data 10 and Fig. 3f, g). From this 215 gene list, which also included miRNAs, we chose 10 genes for further validation. We also included Hey2[38], which functions potentially in early embryonic hemangiogenesis but not ranking high in the screening, and Plcd4, which ranked high in screening but without enriched expression in mesoderm. Out of these, gRNAs for 6 of the 12 genes significantly reduced the ratio of Etv2-tdTomato^high cells to "pr" cells (Fig. 3h and Supplementary Fig. 3b, c). gRNAs against *Flk1* and *Foxh1* exhibited the most significant reduction in Etv2^high cell generation.

**Foxh1 is required for FLK1+ mesoderm generation.** *Foxh1* deficiency leads to early embryonic lethality due to abnormal patterning in gastrulation[36, 37]. Therefore, its direct role in hemangiogenesis has not been revealed. *Foxh1* expression increased gradually during EB development, peaking in "dp" cells (Fig. 4a). Similarly, *Foxh1* expression was higher in "dp" cells compared to "dn" or "pr" cells in E7.5 embryos (Fig. 4b). To investigate the role of *Foxh1* in the Etv2^high cell generation, we generated *Foxh1* knockout TGET and SGET ES cells by utilizing

the CRISPR/Cas9 system (*ΔFoxh1* TGET/*ΔFoxh1* SGET ES cells). *Foxh1* deficiency severely blocked Etv2-tdTomato and FLK1 expression, while T-GFP+ and "pr" cells were still produced, although at somewhat reduced levels (Fig. 4c). Since "pr" cells still developed from *ΔFoxh1* TGET ES cells, we determined if *Foxh1* deficiency leading to Etv2^high cell generation defect was at the level of FLK1+ mesoderm formation from "pr". We established a DOX-inducible *Foxh1* expression system using *ΔFoxh1* ES cells (iFoxh1-*ΔFoxh1* ES cells), which expresses exogenous *Foxh1* in a DOX-inducible manner in the context of endogenous *Foxh1* deficiency (Supplementary Fig. 4a). First, exogenous *Foxh1* expression in *ΔFoxh1* EBs could rescue FLK1+ mesoderm (Supplementary Fig. 4b). We next sorted "pr" cells from D3 Foxh1-rescued *ΔFoxh1* EBs and further cultured them on OP9 cells. While FLK1+ cells were robustly generated in the presence of DOX, very few FLK1+ cells were generated without DOX (Fig. 4d, *middle*, results summarized in Fig. 4e). This suggests that *Foxh1* is essential for directly generating FLK1+ mesoderm from "pr" cells. Moreover, while still able to produce "pr" cells, "dn" cells could not generate FLK1+ cells without DOX. Importantly, almost all "dp" cells even reverted back to "pr" or "dn" stages without DOX, suggesting that *Foxh1* is also required for maintaining the "dp" state. Intriguingly, *Foxh1* overexpression in the context of wild-type background inhibited FLK1+ cell generation (Supplementary Fig. 4c), although this was in line with previous studies in zebrafish[39]. Collectively, these studies suggest a dosage-sensitive *Foxh1* requirement in FLK1+ mesoderm generation.

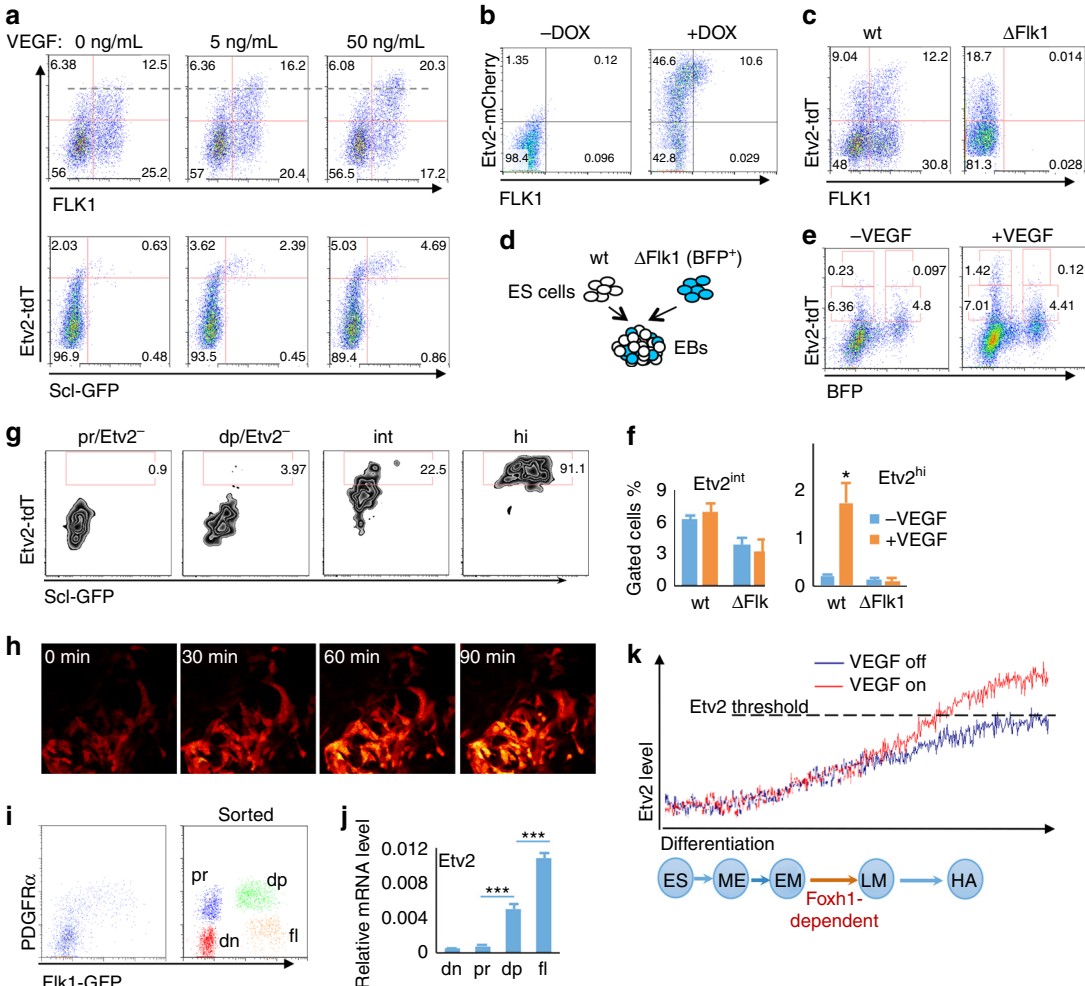

**Fig. 6** VEGF-FLK1 signaling regulates *Etv2* threshold expression. **a** Wild-type (*wt*) SGET ES cells were differentiated in different concentrations of VEGF and analyzed for Etv2-tdTomato and Scl-GFP expression on D4. VEGF was added from D3–4. **b** Flow cytometry analysis of ETV2-mCherry and FLK1 in D3.5 iEtv2-mCherry EB cells differentiated in serum-free conditions. DOX (2 μg/mL) was added from D2.5 to D3.5. **c** Flow cytometry analysis of D4 wt and Δ*Flk1* SGET EB cells for Etv2-tdTomato and FLK1 expression is shown. **d** Scheme of co-differentiation of wt and Δ*Flk1* SGET ES cells is shown. wt and Δ*Flk1* SGET ES cells were mixed at a 4:1 ratio and co-differentiated as described in Experimental procedures. **e** Response of wt and Δ*Flk1* SGET D4 EB cells to VEGF is shown as Etv2[high] cell generation. VEGF (50 ng/mL) was added from D3–4. BFP-negative cells depict wt and BFP-positive cells depict Δ*Flk1* cells. **f** Percentages of Etv2[int] and Etv2[high] cells within wt and Δ*Flk1* SGET EB cells. **g** Sorted populations as indicated from D4 SGET EBs were re-seeded onto OP9 cells and analyzed 12 h later for Etv2-tdTomato and Scl-GFP by flow cytometry. **h** Live imaging of SGET cells responding to VEGF. *Black* regions in the pictures are occupied by Etv2-tdTomato[negative] cells, Etv2-tdTomato[int] cells are in *red* and Etv2-tdTomato[hi] cells are in *orange*. (**i**) FACS plot of D3 Flk1[GFP/GFP] EB cells used for cell sorting, PDGFRα⁻GFP⁻ (*dn*), PDGFRα⁺GFP⁻ (*pr*), PDGFRα⁺GFP⁺ (*dp*), and PDGFRα⁻GFP⁺ (*fl*) cells, is shown. *Etv2* expression in "dn", "pr", and "dp" cells isolated from D3 Flk1[GFP/GFP] EB cells is shown in **j**. **k** A model showing hemangiogenic progenitor development. The *blue circles* indicate different stages during mesoderm differentiation (*ES* undifferentiated ES cells, *ME* mesendoderm, *EM* early mesoderm, *LM* lateral mesoderm, *HA* hemangiogenic progenitors). The zigzag curves at the *top* indicate the stochastic fluctuation of *Etv2* expression. Activation of the hemangiogenic program rarely occurs in the absence of VEGF signaling (*blue line*). However, when the VEGF signaling is active, the hemangiogenic fate can be efficiently achieved through the promotion of *Etv2* threshold expression (*red line*). *P value < 0.05 in Student's *t* test, ***P < 0.001, and n = 3. *Error bars* are s.d

To further understand how *Foxh1* regulates FLK1⁺ mesoderm generation, we compared transcriptome of *Foxh1*-deficient "pr" cells to those of wild-type mesoderm intermediates ("dn", "pr", or "dp") (Fig. 4f and Supplementary Datas 11, 12). Overall, the gene expression pattern of Δ*Foxh1* "pr" cells was close to other mesoderm samples. Most of the nascent/early mesoderm genes enriched in "pr" cells were already upregulated in Δ*Foxh1* "pr" cells, but at a reduced level compared to wild-type "pr" cells (Fig. 4g, Supplementary Fig. 4d, *top*, and Supplementary Data 11). In particular, most of the genes enriched in "dp" cells were more prominently reduced in Δ*Foxh1* "pr" cells compared to wild-type "dp" cells (Supplementary Fig. 4d, *bottom*, and Supplementary Data 11). Among the transcription factors that were extensively expressed in "dp" cells,

*Mixl1*, *Eomes*, *Tbx3*, *Gsc*, and *Mesp1* expression was significantly reduced in Δ*Foxh1* "pr" cells compared to wild-type "dp" cells (Fig. 4h). Therefore, not just the surface marker FLK1, but the specific transcriptional program in FLK1⁺ mesoderm, was blocked by the *Foxh1* deficiency. Consistent with *Foxh1*'s role in gastrulation, gene set enrichment analysis (GSEA)[40] of genes downregulated in Δ*Foxh1* "pr" cells showed significant enrichment of genes that correlate to *T* expression in the early gastrulating embryo[41] (Fig. 4i and Supplementary Fig. 4e). These findings reinforce the notion that *Foxh1* is required for FLK1⁺ mesoderm induction.

**Eomes functions downstream of *Foxh1* in FLK1⁺ cell generation.** Previous studies have shown that *Foxh1* directly activates

expression of nodal, a critical morphogen driving gastrulation and commitment of mesoderm and endoderm[36, 37]. To determine if the defect in FLK1[+] mesoderm generation by *Foxh1* deficiency is cell non-autonomous through nodal production, Δ*Foxh1* TGET ES cells were mixed with wild-type ES cells and differentiated. Wild-type cell supplementation did not rescue FLK1[+] mesoderm from Δ*Foxh1* TGET EBs (Fig. 5a). Therefore, *Foxh1* was intrinsically required for FLK1[+] mesoderm induction.

Although *Foxh1* could rescue FLK1[+] mesoderm, *Flk1* and *Etv2* were not likely to be FOXH1 direct target genes, as upregulation of *Flk1* was observed 24 h after DOX addition, while upregulation of *Etv2* was even later (Fig. 5b–d). To identify potential intermediate gene(s) downstream of *Foxh1*, we analyzed response of "pr-" and "dp"-enriched transcription factors or transcription-related genes that were reduced in Δ*Foxh1* "pr" cells (Fig. 4g and Supplementary Data 11), and became rescued by exogenous *Foxh1* expression (Fig. 5c). Among the genes showing upregulation after 8 h of DOX treatment, *Eomes* is a target for FOXH1[42, 43], and is required for normal embryonic mesoderm development[44, 45]. As *Eomes* expression was enriched in "dp" cells from EBs and E7.5 embryos (Fig. 5e, f), we determined if *Eomes* functions also downstream of *Foxh1* in FLK1[+] mesoderm generation from "pr" nascent mesoderm. Thus, we established *Eomes*-inducible ES cell line in the background of *Foxh1* knockout (iEomes-Δ*Foxh1*). As shown, exogenous *Eomes* expression was able to rescue FLK1[+] cells, although not fully, from *Foxh1*-deficient cells (Fig. 5g). After *Eomes* induction, *Etv2* was also upregulated (Fig. 5h). Collectively, *Eomes* functions downstream of *Foxh1* in FLK1[+] mesoderm generation.

**VEGF-FLK1 signaling regulates *Etv2* threshold expression**. The finding that the VEGF receptor *Flk1* ranked on top among genes required for generating Etv2-tdTomato[high] cells suggested that FLK1 functions upstream of *Etv2*. This notion is consistent with previous studies, which demonstrated that VEGF was able to activate *Etv2* expression[12, 46]. However, *Flk1* knockout embryos or ES cells can still generate hematopoietic progenitors, although limited[35, 47, 48]. This suggests that *Etv2*-mediated hemangiogenic program commitment may occur independently of VEGF-FLK1 signaling. Intriguingly, Etv2-tdTomato[high] cells were noticeably increased in the FLK1[+] population in response to VEGF (Fig. 6a, top). Enhanced generation of Etv2-tdTomato[high] cells by VEGF in the FLK1[+] cells led to more Scl-GFP[+] cells (Fig. 6a, bottom). Meanwhile, Etv2[high] cells reinforced FLK1 levels (Fig. 6a, b), consistent with previous studies that ETV2 can directly activate *Flk1*[8, 10, 23]. To better elucidate the relationship between *Etv2* and *Flk1*, we generated *Flk1* knockout SGET ES cells using the CRISPR/Cas9 system (Δ*Flk1* SGET ES cells). As shown, Δ*Flk1* ES cells efficiently expressed *Etv2*, suggesting an independent regulation of *Etv2* from *Flk1* (Fig. 6c). However, Etv2-tdTomato[high] cells were predominantly found in FLK1[+] cells and they were specifically missing in Δ*Flk1* cells (Fig. 6c). Moreover, when mixed and co-differentiated together with wild-type ES cells (BFP negative) (Fig. 6d), Δ*Flk1* ES cells (BFP positive) very rarely generated Etv2-tdTomato[high] cells, even though Etv2-tdTomato[int] cells were readily observed (Fig. 6e, left). While wild-type cells, i.e., BFP negative, were significantly elevated into the Etv2[high] state by VEGF, Δ*Flk1* cells, i.e., BFP positive, showed no response to VEGF (Fig. 6e, right, summarized in Fig. 6f). Consistently, SGET cells infected with lentivirus expressing gRNA against *Flk1* (BFP positive) produced significantly reduced levels of Scl-GFP[+] cells compared to un-infected cells (Supplementary Fig. 5a). Therefore, VEGF-FLK1 signaling is required for Etv2[high] cell generation, not for the initiation of *Etv2* expression, and a reciprocal promotion between FLK1 and *Etv2* might maintain the cells in the Etv2-tdTomato[high] state.

We next determined if VEGF signaling directly promotes Etv2[high] cells from Etv2[int] cells. Compared to Etv2-tdTomato[−] "pr" and Etv2-tdTomato[−] "dp" cells, Etv2-tdTomato[int] cells, when re-seeded onto OP9 cells, could generate more Etv2-tdTomato[high] cells (Fig. 6g). Furthermore, we observed direct elevation of tdTomato expression in Etv2-tdTomato[int] cells into Etv2-tdTomato[high] cells in the presence of VEGF (Fig. 6h). To further confirm VEGF's role in *Etv2* expression, we generated *Vegfa* knockout SGET cells using the CRISPR/Cas9 system (Δ*Vegfa* SGET ES cells). We induced mesoderm differentiation with BMP4 in serum-free conditions[33] (Supplementary Fig. 5b, c). In this assay, Δ*Vegfa* cells reproduced the phenotype of Δ*Flk1* cells, while a supplement of exogenous VEGF rescued the generation of Etv2-tdTomato[high] and Scl-GFP[+] cells. We noticed that even without VEGFA supplementation, *Etv2* was still efficiently expressed in FLK1[+] cells, although not efficiently achieving the threshold expression (Supplementary Fig. 5c), again demonstrating the VEGF role in the generation of Etv2-tdTomato[high] cells.

To explore the dynamics of VEGF-FLK1 signaling-independent *Etv2* expression, we utilized the Flk1[GFP/GFP] ES cell line[49]. This cell line in differentiation showed a similar coexpression pattern of PDGFRα and GFP as PDGFRα and FLK1 in wild-type cells, indicating that GFP expression faithfully reflected that of *Flk1* (Fig. 6i). As expected, very few GFP single positive cells, in which *Etv2* expression and hemangiogenic progenitors were enriched[10, 23], were generated from Flk1[GFP/GFP] ES cells (Fig. 6i). We sorted PDGFRα and GFP (FLK1) double negative cells ("dn"), PDGFRα single positive cells ("pr"), PDGFRα and GFP double positive cells ("dp"), and GFP single positive cells ("fl"), and measured *Etv2* expression by qRT-PCR. Remarkably, *Etv2* expression was readily upregulated in PDGFRα[+]GFP[+] cells compared to "pr" cells, similar to that in PDGFRα[+]FLK1[+] cells, suggesting VEGF-FLK1 signaling-independent *Etv2* expression was initiated mainly in the presumptive FLK1[+] mesoderm (GFP[+] cells) (Fig. 6j). Very rare GFP single positive cells expressing higher level *Etv2* can still be made, probably through stochastic mechanisms. Collectively, FLK1[+] mesoderm is enriched for *Etv2* basal level expression and VEGF-FLK1 signaling instructs hemangiogenic fate specification by ensuring *Etv2* threshold expression.

**Discussion**

In this study, we provide mechanisms by which hemangiogenic lineage commitment is achieved. In particular, by tracking *Etv2* and *Scl* reporter expression and by comparing transcriptome of various mesodermal cell populations, we were able to delineate the developmental route generating hematopoietic and endothelial cells via T/Bry[+]PDGFRα[−]FLK1[−], T/Bry[+]PDGFRα[+]FLK1[−], PDGFRα[+]FLK1[+], FLK1[+]Etv2[int], FLK1[+]Etv2[high], to Etv2[high]Scl[+] cells. In this process, achievement of *Etv2* threshold expression is key to the hemangiogneic lineage specification. Currently, FLK1 expression is widely used as a hemangiogenic lineage marker[28, 41, 50]. However, the highly refined lineage route established in this study indicates that hemangiogenic progenitors represent a small fraction of FLK1[+] cells. Therefore, in the future, this improved lineage map should be helpful to isolate relevant cell populations to further define gene regulatory network involved in hematopoietic and vascular lineage development.

PDGFRα has been extensively used as paraxial mesoderm marker[20, 51]. In particular, Sakurai et al. have suggested that FLK1[+]PDGFRα[+] mesoderm represents nascent mesoderm, from which both FLK1[+] lateral plate mesoderm and PDGFRα[+] paraxial

mesoderm are generated[20]. However, they also established that "dn", "pr", and "dp" cells are developmentally plastic and interchangeable in culture. Moreover, PDGFRα expression marks nascent mesoderm in gastrulating mouse embryos[52]. Our transcriptome analysis revealed that "pr" cells are close to $T^+$ PDGFRα$^-$FLK1$^-$ mesendoderm and the proximal nascent mesoderm of the primitive streak of the E7.0 gastrulating embryo[27]. Furthermore, FLK1$^+$PDGFRα$^+$ mesoderm has cardiac potential[19, 23]. These data support that "pr" cells represent nascent/early mesoderm from which "dp" cells arise. Consistently, we found that loss of Foxh1 or treatment with the ERK inhibitor blocked progression of "pr" cells to "dp" state. Potentially, later arising "pr" cells could include PDGFRα$^+$ paraxial mesoderm and cardiac progenitors. However, we cannot rule out the possibility that FLK1$^+$PDGFRα$^+$ cells can also be generated directly from mesendoderm when optimal stimuli for "dn" and "dp" stages are simultaneously available. Future studies are warranted to show the in vivo relationship between PDGFRα$^+$ cells and FLK1$^+$ cells.

Currently, upstream signals that activate Etv2 have not been clearly elucidated. It appears that ETV2 positively regulates its own expression[10, 53]. The cAMP-PKA-CREB, Calcineurin-NFAT, MESP1-CREB1, VEGF-p38 MAPK-CREB, as well as NKX2-5 have been suggested to activate Etv2[10, 46, 53–56]. In this paper, we applied CRISPR/Cas9 genome-wide screening to developmental processes to identify Foxh1 and Flk1 to be upstream of Etv2 expression. By utilizing in vitro differentiation of ES cell model, we demonstrated that Foxh1 deficiency leads to a block in FLK1$^+$ mesoderm generation from early "pr" mesoderm. These studies revealed an unexpected role of Foxh1 beyond the anterior-posterior patterning in gastrulation. We demonstrated the potential role of Eomes, one FOXH1 target gene, in FLK1$^+$ mesoderm generation. Recently, npas4l gene was reported to be upstream of etv2 in zebrafish[57]. Future studies delineating upstream pathways that control Etv2 threshold expression are warranted.

Once achieving the Etv2 threshold expression and the hemangioblast specification, ETV2 may initiate a network critical for hematopoietic and/or endothelial generation, by activating key hematopoietic and endothelial cell factors and establishing the Ets hierarchy[10]. In particular, upon activation by ETV2, Scl establishes a transcription factor network that forms a positive feedback loop, consistent with previous observations[16]. Given the critical function of Scl in hematopoietic development, it would be informative to understand the essential structure of this positive feedback network. Recently, Goode et al. proposed a core regulatory network model for hematopoietic specification[28]. Almost all these core transcription factors, which are direct targets of ETV2[10], co-occupy and regulate a large set of hematopoietic genes. Among these, coexpression of only SCL and LMO2 was sufficient to reprogram fibroblasts into hematopoietic cells[28]. It should be determined in the future whether the same group of transcription factors can establish the SCL-positive feedback loop in the absence of ETV2. Previous studies have reported that ETV2 and/or other ETS factors can cooperate with FOXC factors to activate endothelial genes, implying that ETS and FOXC factors may form an endothelial autoregulatory network[58]. Future studies will require identifying a minimal network(s) that is (are) necessary for hematopoietic and endothelial cell generation.

The unexpected findings of this study were that Etv2 expression is initiated independently of FLK1 or VEGF and that VEGF plays an instructive role in the hemangiogenic fate determination by conferring Etv2 threshold expression. Accordingly, a potential positive feedback interaction between Etv2 and FLK1 would ultimately generate the Etv2$^{high}$FLK1$^{high}$ state. Cells that reach the Etv2$^{high}$FLK1$^{high}$ state are now committed to the hemangiogenic fate. Therefore, Vegfa, Flk1, and Etv2 form a key regulatory module in the hemangiogenic fate commitment. Indeed, genetic knockout studies have established that Vegfa, Flk1, or Etv2 deficiency leads to similar hematopoietic and endothelial cell formation defects[8, 35, 59, 60]. In particular, only FLK1$^{high}$ cells were missing in Etv2-deficient embryos[10]. Moreover, Etv2 deletion using the Flk1-Cre system results in embryonic lethality, apparently showing decreased generation of the FLK1$^{high}$PDGFRα$^-$ cell population[10]. This suggests that any perturbation of the Etv2, VEGF, and FLK1 module causing insufficient generation of Etv2$^{high}$FLK1$^{high}$ hemangiogenic progenitors might lead to defects in hematopoietic and endothelial cell development. Collectively, we propose a two-step model for the hemangiogenic fate commitment: VEGF-FLK1 signaling-independent Etv2 initiation and VEGF-FLK1 signaling-dependent Etv2 threshold expression (Fig. 6k). Very rarely, Etv2 threshold expression may also be achieved through intracellular stochastic mechanisms (as depicted in zigzag lines in Fig. 6k). In conclusion, VEGF signaling plays an instructive role in hemangiogenesis, as it enforces new fate acquisition, i.e., hemagniogneic fate, through promoting Etv2 threshold expression.

## Methods

**Generation of ES lines.** All mouse ES cells were maintained on mouse embryo fibroblast (MEF) feeder cell layers in Dulbecco-modified Eagle medium containing 15% fetal bovine serum, 100 units/mL LIF, 1× MEM Non-Essential Amino Acids Solution (Gibco), 1× Glutamax$^{TM}$ Supplement (Gibco), and $4.5 \times 10^{-4}$ M 1-Thioglycerol (MTG, Sigma). For differentiation, ES cells grown on MEFs were split onto a gelatin-coated dish in Iscove's modified Dulbecco medium (IMDM) with the same supplements as used for maintenance, and were cultured for 2 days. Single-cell suspensions were then prepared, and 4000–10,000 ES cells were added per mL to a differentiation medium of IMDM containing 15% differentiation-screened fetal calf serum, 1× Glutamax, 50 μg/mL ascorbic acid, and $4.5 \times 10-4$ M MTG on a bacteriological Petri dish. For serum-free differentiation, FCS was replaced with 15% knockout SR (Gibco). OP9 cells were grown in α-MEM supplemented with 20% differentiation-screened FCS, 1× Glutamax, 1× NEAA, and $4.5 \times 10-4$ M MTG[8], and sorted EB cells growing on OP9 cells were in the same medium. For ERK inhibition, PD0325901 (Sigma) was added at 2 μM. For serum-free differentiation of ΔVegfa SGET ES cells, 5% PFHM II (Gibco) was supplemented, and mouse BMP4 (BDbiosciences) and VEGF-A (Pepro Tech) were added as indicated. SGET ES cells were generated by utilizing Scl-EGFP knock-in ES cells[17] and inserting the tdTomato coding sequence after the ETV2 stop codon using the T2A. Briefly, donor vector containing 1.9 kb upstream (5′-arm) and 2.4 kb downstream (3′-arm) of the ETV2 stop codon were cloned into the pGolden-Hyg plasmid (Addgene). The stop codon was replaced with the T2A-tdTomato coding sequence. The sgRNA sequence directing a cut at 13 bp upstream of the ETV2 stop codon was inserted into the CRISPR plasmid PX330 (Addgene, deposited by Feng Zhang's lab). Donor vector and CRISPR plasmid were cotransfected into Scl-EGFP knock-in ES cells using the Lonza nucleofection kit. Hygromycin was added 24 h later (200 μg/mL, Sigma) and selected for 7 days. Hygromycin-resistant clones were picked and identified by PCR and sequencing. The hygromycin-resistance cassette was subsequently excised by nucleofection of the cells with Cre[61]. The sgRNA sequence for targeting Etv2 locus and primers used to identify the correctly targeted clones are listed in Supplementary Table 1. TGET ES cells were generated by utilizing Brachyury/EGFP knock-in ES cells[18] and the same strategy for generating SGET cells. Inducible ES cell lines (iEtv2-mCherry, iFoxh1, and iEomes) were generated by targeting the tet-responsive locus of A2Lox/Scl-hCD4 cells[61], with the construct containing the coding sequence of Etv2-mCherry, Foxh1, or Eomes, as described previously. For generating homozygous/heterozygous knockout ES cell lines, ES cells were first transduced with lenti-Cas9-Blast (Addgene, deposited by Feng Zhang Lab) and selected using blasticidin (Sigma, 2 μg/mL). Single clones were picked for identification of Cas9 expression and normal differentiation. sgRNAs were cloned into lentiGuide-Puro (Feng Zhang Lab) or pKLV-U6gRNA_BbsI_PGKPuro2ABFP[62] that coexpresses blue fluorescent protein (BFP) with the sgRNA. The cells were infected with lentiviruses expressing sgRNA against the target gene. The cells were selected with puromycin (Sigma, 1 μg/mL) for at least 7 days and then single clones were picked and identified by sequencing. The sgRNAs used to identify the mutations are listed in the Supplementary Table 1.

**Flow cytometry and sorting.** EBs were dissociated with Accutase solution (Sigma). E7.5 embryos were first digested in 0.25% collagenase and 20% FBS, then in accutase solution. Single cells were incubated with α-human CD4 (Biolegend), α-mouse FLK1 (BioLegend or eBioscience), α-mouse PDGFRα (BioLegend),

α-mouse CD41 (eBioscience) or α-mouse TIE2 (eBioscience). Data were acquired on the FACS Canto II or LSR-Fortessa flow cytometer (BDbiosciences) and analyzed using the FlowJo (Treestar) software. Cell sorting was performed using the BD FACSAria II system.

**Blast colonies formation cell assay**. Sorted EB cells were replated in methyl cellulose containing 10% plasma-derived serum (PDS, Antech; Texas), 5% PFHM II, 1× Glutamax, 1× BIT9500 (STEMCELL Technologies) and MTG (4.5 × 10−4 M), together with kit ligand (1% conditioned medium). Blast colonies were counted 4 days later[15].

**RNA-seq analysis**. Total RNA was extracted using the RNeasy Mini kit (Qiagen). Library preparation was performed with 10 ng of total RNA (GTAC core facility, Washington University in St. Louis): Total RNA integrity was validated using an Agilent bioanalyzer. All samples were prepared using the Ribozero (Epicentre) kit as per the manufacturer's protocol. cDNA was then blunt ended, an A base added to the 3′ ends, and then Illumina sequencing adapters were ligated to the ends. Ligated fragments were then amplified for 12 cycles using primers incorporating unique index tags. Fragments were sequenced on an Illumina HiSeq-2500 using single reads extending 50 bases. RNA-seq reads were aligned to the mouse mm9 assembly from the UCSC Genome Browser with Tophat2[63]. Gene counts were derived from the number of uniquely aligned unambiguous reads by Subread: featureCount version 1.4.5[64]. All gene-level counts were then imported into the R/Bioconductor package EdgeR normalized to adjust for differences in library size. Genes with RPKM <5 in all samples were excluded from further analysis. Genes with of less than fivefold changes in any two samples were also excluded. Finally, 3815 genes were used for further presentation. Published data used in RNA-seq analysis: GSE57409 (sample "ES" and "EpiSC"), GSE36114 (sample "endo_d6", representing ES cell for endoderm differentiation on day 6), GSE69080 ("mes", "hb", "hp"), GSE55310 ("he"), 7R2 (https://b2b.hci.utah.edu/gnomex/, sample "cp", representing ES cells differentiated to cardiac progenitors[25]). The RNA-seq data have been deposited in the NCBI Gene Expression Omnibus (GEO) database under accession code GSE85641.

**Zipcode mapping**. Zipcode mapping was performed as described in the developers' manual[27] (www.itranscriptome.org/). Briefly, after uploading transcriptome data of our samples to the website, expression values of 158 preselected landmark genes of the E7.0 gastrulating embryo were extracted. Then a matrix of Spearman rank correlation coefficients of the transcriptome of a given sample to those of different regions in the gastrulating embryo was returned from the website.

**CRISPR genetic screening and data analysis**. CRISPR screening was performed as previously described[11]. Briefly, TGET ES cells were stably transduced with lenti-Cas9-Blast and selected using blasticidin (2 ng/mL). Subsequently, 2.4 × 10⁷ TGET ES cells constitutively expressing Cas9 were infected with the GeCKO CRISPR lentivirus library at a MOI of 0.2 (200× coverage of the library). Cells were selected with puromycin and maintained in ES medium for 2 weeks, followed by differentiation and cell sorting. Cell numbers in each sample for genomic DNA extraction ensured at least 100× coverage of the library. Genomic DNA extraction and sgRNA library amplification were performed faithfully following the previously described protocol[11]. Sequencing data preprocess was also performed as described in the same protocol. The read counts of each sgRNA sequence were imported to HiTSelect[30] to obtain gene lists with significance of sgRNA depletion in treated samples compared to control samples. Functional annotation of gene lists was performed with ConsensusPathDB (http://cpdb.molgen.mpg.de/MCPDB). The CRISPR screen reads data have been deposited in the NCBI Gene Expression Omnibus (GEO) database under accession code GSE85641. DNA sequences for primers and sgRNAs listed in Supplementary Table 1.

**Gene set enrichment analysis**. Gene set enrichment analysis (GSEA) analysis was carried out using the GSEA software from the Broad Institute[40]. Genes were ranked by the downregulation fold in Foxh1 knockout "pr" cells compared to wt "pr" cells. Genes with expression correlated to T in early gastrulation embryo[41] (correlation score >0.2) were used as the test gene sets for analysis of enrichment.

**Quantitative real-time reverse transcription PCR**. Total RNA from embryos or EB cells was prepared with RNeasy Micro/Mini Kit (Qiagen), and reverse transcribed into cDNAs according to the manufacturer's protocol. Expression of genes was measured by quantitative real-time RT-PCR using primers indicated in Supplementary Table 1. Gene expression levels were normalized to Gapdh.

**Live imaging**. SGET D3 EBs were transferred onto Matrigel-coated glass bottom dish for 24 h, followed by live imaging. VEGF was added at 50 ng/mL and the cells were maintained in a 5% CO₂ wet chamber fitted onto the Nikon a1 confocal microscope. Live images were captured every 30 min.

**Statistical analysis**. The results of qRT-PCR and flow cytometry analysis were analyzed by Students' t test. P < 0.05 was considered significant.

**Data availability**. The authors declare that all data supporting the findings of this study are available within the article and its supplementary information files or from the corresponding author on reasonable request. The RNA-seq data and CRISPR screen reads counting data reported in this paper have been deposited in NCBI GEO under accession code GSE85641.

Published data used in RNA-seq analysis include the following already deposited data sets: GSE57409 (sample "ES" and "EpiSC"), GSE36114 (sample "endo_d6", representing ES cell for endoderm differentiation on day 6), GSE69080 ("mes", "hb", "hp"), GSE55310 ("he"), 7R2 (https://b2b.hci.utah.edu/gnomex/, sample "cp", representing ES cells differentiated to cardiac progenitors[25]).

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

## Acknowledgements

We thank the lab members for constructive criticism and helpful discussion. We thank Karen Krchma for help with mouse work, Rong Zhang for help with CRISPR lentivirus library preparation, and Gordon Keller for T/EGFP ES cells. We thank Masa Ema for Flk1GFP/GFP ES cells. We thank the Genome Technology Access Center at Washington University School of Medicine for deep sequencing service. This work was supported by NIH Grants R01HL63736 and R01HL55337 (K.C.).

## Author contributions

H.Z. and K.C. conceived and designed experiments, interpreted data, and wrote the paper. H.Z. performed experiments and analyzed data.

## Additional information

**Competing interests:** The authors declare no competing financial interests.

