## [Peer Review File · Nature Communications]

Reviewers' Comments:

Reviewer #1 (Remarks to the Author)

This manuscript by Zhao and Choi explores the upstream control of Etv2 expression and further defines the relationship between ETV2 and the FLK1-VEGF signalling axis. Through a series of experiments using gain-and-loss of functions the authors demonstrate the tight and reciprocal regulation between ETV2 and FLK1 and the need for a threshold level for ETV2 to specify hemangioblast fate. This manuscript also describes a CRISPR screen approach to uncover upstream regulators of ETV2, which led to the identification of FOXH1 and its functional characterization through loss of function. Overall, this is an interesting manuscript, which presents convincing evidences to support the main claims and conclusions.

However, this study would be strengthened by the addition of in vivo data linking what is observed in the in vitro ESC system to the developing embryo, such as expression pattern for FOXH1, detection of differential expression levels for ETV2 in the developing embryo. Furthermore, throughout the manuscript the presence or absence of hemangioblast is only characterized by flow cytometry, it would be important to perform clonogenic assay for this progenitor to further validate the flow data presented. Finally, while the discovery of FOXH1 as an upstream regulator of Etv2 is certainly interesting, it would be important to establish whether Etv2 is a direct transcriptional target of FOXH1 or whether FOXH1 is required for the generation of a FLK1+ population in which Etv2 is expressed.

Additional comments:

Why are the two alleles of Etv2 modified with the 2A-tdTomato? Is the difference between intermediate and high Etv2-tdT populations a difference between one or two alleles being expressed?

Does the enforced expression of Etv2-mCherry results in similar Etv2 transcription level when compared to the wild type situation? It is surprising that this inducible system results in the generation of populations with intermediate and high Etv2 levels.

The transition from PDGFR+FLK1- to PDGFR+FLK1+ (as shown in fig 1f-h) needs to be formally demonstrates.

Does the ETV2^{int} population gives rise to ETV2^{high} cells or do they emerge separately Brachyury+ mesoderm and hence represent different mesoderm subsets?

It is not clear how data in figure sup 1h were obtained. "Transcriptome of pr was compared to those of 11 tandem regions along the primitive streak of E7.0 mouse gastrulating embryo, using Zipcode mapping¹⁴." It does not seem that reference 14 explain Zipcode mapping or contains data from the 11 tandem regions.

Page 5-line 141: what do the authors refer to in the sentence "Having established that Etv2 cells develop through a distinct developmental route"? What is this developmental route distinct from?

It would be important to present real time PCR validation for Foxh1 expression in the various subpopulation of differentiating ESCs.

Supplemental figure 4a does not seem to be described in the text.

Reviewer #2 (Remarks to the Author)

Zhao and Choi report a comprehensive investigation of the role of key transcription factors and signalling molecules during the early process of blood specification from mesoderm in mouse ESC differentiation. The study nicely complements gene targeting (KO and reporter lines) with differentiation assays, expression profiling and a CrispR KO screen. There are some points where I feel the paper could be improved, as outlined below.

Specific Comments:

1) The expression data should be compared to last year's paper by Goode et al (Dev Cell), because this previous paper reported expression as well as ChIP-Seq datasets. It will be important for the wider community to know how similar (or not) some of the populations are. Moreover, the Goode dataset can be exploited to build hypotheses on regulatory mechanisms since it also includes ChIP-Seq TF binding maps.

2) Scialdone et al (Nature 2016) reported genes associated with the initiation of gastrulation in mouse embryos. It will be important to see the degree of overlap with the Foxh1 KO data reported in this paper here, given the known role of Foxh1 in early embryo patterning.

3) I could not access the GEO accession number in the paper. It will be important for this to be made public if the paper gets accepted. I would have liked to see how they submitted their CrispR screen counts. I realize that the paper contains supplementary tables but are these also attached to the GEO submission (e.g. are there some processed data for the CrispR sequence reads; it would be useful if there are).

4) It would be good to provide a little more discussion of the previously described interplay between Ets and Fox transcription factors (for example work by Sarah de Val). It seems to me that there could be feed forward loops going on, which could be a nice speculation.

Minor

Line 94: hemangiogenic is misspelled

We want to thank the reviewers for the positive comments and valuable suggestions on the manuscript. The reviewers expressed some concerns on the manuscript. Accordingly, we performed new experiments and modified the manuscript as best as we could to accommodate reviewers' comments. Particularly, inclusion of the embryo data and comparison of our data with published data further strengthened our major arguments. We additionally explored the mechanisms for *Foxh1* regulating FLK1⁺ mesoderm generation. We found that *Eomes*, a *Foxh1* target gene, at least partially mediates *Foxh1*'s function in FLK1⁺ mesoderm generation. We hope the changes are now acceptable to the reviewers. Figures 1b-right, 2e, 2g, 4d, 4i, 5b-h, 6g-h, and Supplemental Figures 1b, 1e, 2b, 2d-e, 4e are new. Sections containing changes are marked in red. Also, this revised manuscript now includes separate introduction and discussion sections. Our response to each comment is as follows.

Reviewer #1 (Remarks to the Author):

This manuscript by Zhao and Choi explores the upstream control of Etv2 expression and further defines the relationship between ETV2 and the FLK1-VEGF signalling axis. Through a series of experiments using gain-and-loss of functions the authors demonstrate the tight and reciprocal regulation between ETV2 and FLK1 and the need for a threshold level for ETV2 to specify hemangioblast fate. This manuscript also describes a CRISPR screen approach to uncover upstream regulators of ETV2, which led to the identification of FOXH1 and its functional characterization through loss of function. Overall, this is an interesting manuscript, which presents convincing evidences to support the main claims and conclusions.

1) However, this study would be strengthened by the addition of in vivo data linking what is observed in the in vitro ESC system to the developing embryo, such as expression pattern for FOXH1, detection of differential expression levels for ETV2 in the developing embryo.

We sorted E7.5 embryos into PDGFR α -FLK1⁻ ("dn"), PDGFR α single positive ("pr"), PDGFR α and FLK1 double positive ("dp"), and FLK1 single positive ("fl") cells and analyzed mRNA levels of *Etv2*, *Oct4*, *Foxh1*, and *Foxa2*. This new data is now shown in Fig. 2g and 4b, which demonstrates that the gene expression pattern from E7.5 embryo derived cell populations was consistent with that from EB derived cell populations.

2) Furthermore, throughout the manuscript the presence or absence of hemangioblast is only characterized by flow cytometry, it would be important to perform clonogenic assay for this progenitor to further validate the flow data presented.

Blast colony assay, which is used as the hemangioblast readout¹, was carried out to show that *Etv2*-tdTomato^{high} cells are indeed enriched for hemangioblasts. As presented, ~2% (Fig. 1b, right) - ~4% (Fig. 2e) of the replated *Etv2*-tdTomato^{high} cells generated blast colonies. Although this frequency seems low, we attribute this to the fact that blast colony assay is very sensitive to the cell density and single cell manipulations, in line with previous findings that ~1% of sorted FLK1⁺Scl⁺ cells formed blast colonies². Please note, however, that Fig. 1b-right was performed at sub-optimal cell density and that Fig. 2e, which was performed at higher cell density, does not have replicates, as we could not sort sufficient number of *Etv2*-tdTomato^{high} cells due to their low abundance. Collectively, clonogenic assay validates the notion that *Etv2*-tdTomato^{high} cells are enriched for hemangiogenic (blast-colony forming cell) progenitors.

3) Finally, while the discovery of FOXH1 as an upstream regulator of Etv2 is certainly interesting, it would be important to establish whether Etv2 is a direct transcriptional target of

FOXH1 or whether FOXH1 is required for the generation of a FLK1+ population in which Etv2 is expressed.

Our data indicated that *Foxh1* is required for the generation and maintenance of FLK1⁺ mesoderm. To determine if this effect was direct, we measured mRNA dynamics of a list of genes after enforced *Foxh1* expression (+DOX). As shown in Fig. 5 b-d, *Gsc* and *Eomes*, known FOXH1 target genes, were readily upregulated 8 hours after *Foxh1* expression (Fig. 5b, c). However, upregulation of *Flk1* and *Etv2* expression occurred much later following DOX treatment. This suggests that *Flk1* and *Etv2* may not be direct targets of FOXH1, although it is still possible, given that many key developmental genes show dosage effect, that DOX mediated *Foxh1*-rescuing expression level may not be optimal for *Flk1* and/or *Etv2* expression.

To further explore the mechanisms by which FOXH1 regulates FLK1⁺ mesoderm generation, we analyzed "pr" and "dp" enriched transcription factors or transcription-related genes that were reduced in *Foxh1*-deficient "pr" cells, but became rescued by exogenous *Foxh1* expression (Fig. 5c). Among those that showed early response to *Foxh1* rescuing (8 hours), *Eomes* has been reported to be important for mesoderm development³. To determine if *Eomes* could function downstream of *Foxh1* in inducing FLK1⁺ mesoderm from "pr" nascent mesoderm, we established an ES cell line with inducible *Eomes* expression in the background of *Foxh1*-knockout. As shown in Fig. 5g, exogenous *Eomes* could partially rescue *Foxh1* deficiency in FLK1⁺ cell generation. Collectively, we conclude that *Foxh1* mediated *Eomes* induction is integral to FLK1⁺ mesoderm generation.

4) Why are the two alleles of Etv2 modified with the 2A-tdTomato? Is the difference between intermediate and high Etv2-tdT populations a difference between one or two alleles being expressed?

We did not willingly modify one vs two alleles of the *Etv2* locus when we introduced the 2a-tdTomato using the CRISPR technology. We happened to obtain two ES cell lines, one with only one allele carrying the reporter (in TGET cells), while the other cell line (SGET) has both alleles carrying the reporter. Whether tdTomato expression is from one allele or both, we find that *Etv2* threshold expression, i.e. tdTomato expression levels, was similar.

This reviewer was possibly suggesting a very attractive idea that the hemangiogenic fate might be specified via stepwise activation of the 2 *Etv2* alleles. However, first, *Etv2*^{+/-} mice don't have any developmental defects in hematopoietic and vascular systems⁴, and second, when we followed tdTomato expression in *Etv2* heterozygous SGET cells, we observed similar threshold level requirements in activating *Scf* expression (Supplementary Fig. 1b). Therefore, it is more likely that *Etv2* threshold requirement is for the total ETV2 protein levels.

5) Does the enforced expression of Etv2-mCherry results in similar Etv2 transcription level when compared to the wild type situation? It is surprising that this inducible system results in the generation of populations with intermediate and high Etv2 levels.

It is important to note that mCherry was fused to *Etv2* to generate *Etv2*-mCherry protein. *Etv2*-mCherry fusion protein likely behaves similar to the endogenous protein, potentially resulting in the detection of *Etv2*^{int} and *Etv2*^{high} cells. When we compared *Etv2* mRNA levels of *Etv2*^{high}*Scf*⁻ cells sorted from SGET EBs generated in serum or *Etv2*-mCherry^{high}*Scf*⁻ cells from i*Etv2*-mCherry EBs generated in serum-free conditions (Supplementary Fig. 1e), *Etv2*

expression levels from Etv2-mCherry^{high} cells were higher compared to those from Etv2-tdTomato^{high} cells. This result suggested that the exact threshold level of Etv2 expression for activating Scf in different systems (such as serum vs serum-free) may be different.

6) The transition from PDGFR⁺FLK1⁻ to PDGFR⁺FLK1⁺ (as shown in fig 1f-h) needs to be formally demonstrates.

Based on gene expression patterns in Figure 2 and 4, we argue that PDGFR α is in fact the maker of nascent mesoderm that generates FLK1 positive lateral mesoderm. Consistently, inhibition of the ERK activity in D2.5 EBs blocked specification of FLK1 positive cells from PDGFR α ⁺FLK1⁻ cells (Fig. 2c). Furthermore, in early EBs PDGFR α ⁺FLK1⁻ cells have closer gene expression pattern to T/Bry⁺PDGFR α ⁻FLK1⁻ mesendoderm than PDGFR α ⁺FLK1⁺ cells, while in E7.5 embryos PDGFR α ⁺FLK1⁻ cells have gene expression pattern close to PDGFR α ⁻FLK1⁻ cells (Fig. 2f, g, Supplementary Fig. 2e). Moreover, “dp” cells can be rescued from Foxh1-deficient “pr” by exogenous Foxh1. Thus, it is likely that PDGFR α ⁺FLK1⁻ cells progress to PDGFR α ⁺FLK1⁺ cells. However, it is important to point out that as we discussed in the text, T/Bry⁺PDGFR α ⁻FLK1⁻ (T⁺-“dn”), PDGFR α ⁺FLK1⁻ (“pr”) and PDGFR α ⁺FLK1⁺ (“dp”) cells are developmentally plastic. We cannot rule out the possibility that PDGFR α ⁺FLK1⁺ cells can be generated directly from mesendoderm when optimal stimuli for “dn” and “dp” stages are simultaneously available as well.

7) Does the ETV2^{int} population gives rise to ETV2^{high} cells or do they emerge separately Brachyury⁺ mesoderm and hence represent different mesoderm subsets?

We performed two independent studies to determine the developmental relationship between the Etv2^{int} and Etv2^{high} cells. First, we sorted ETV2^{int} cells and reseeded them onto OP9 cells. Twelve hours later, Etv2^{int} cells generated more ETV2^{high} cells than Etv2 negative, “pr” cells, or Etv2 negative “dp” cells (Fig. 6g). More importantly, we observed direct emergence of ETV2-tdTomato^{high} cells from ETV2-tdTomato^{int} cells using live cell imaging (Fig. 6h). These data demonstrate that Etv2^{int} cells progress to Etv2^{high} hemangiogenic progenitors.

8) It is not clear how data in figure sup 1h were obtained. "Transcriptome of pr was compared to those of 11 tandem regions along the primitive streak of E7.0 mouse gastrulating embryo, using Zipcode mapping¹⁴." It does not seem that reference 14 explain Zipcode mapping or contains data from the 11 tandem regions.

Zipcode mapping is described in supplemental methods.

9) Page 5-line 141: what do the authors refer to in the sentence "Having established that Etv2 cells develop through a distinct developmental route"? What is this developmental route distinct from?

We did not mean to argue that the developmental pathway we identified is distinct from any other previous studies. We meant to say that Etv2 cells develop through a defined developmental route. We have corrected the statement to reflect this.

10) It would be important to present real time PCR validation for Foxh1 expression in the various subpopulation of differentiating ESCs.

The results are now shown in Supplemental figure 4a.

11) Supplemental figure 4a does not seem to be described in the text.

The Figure is now shown as Fig. 6b and described in the main text to support the reciprocal regulation between *Etv2* and *Flk1*.

Reviewer #2 (Remarks to the Author):

1) The expression data should be compared to last year's paper by Goode et al (Dev Cell), because this previous paper reported expression as well as ChIP-Seq datasets. It will be important for the wider community to know how similar (or not) some of the populations are. Moreover, the Goode dataset can be exploited to build hypotheses on regulatory mechanisms since it also includes ChIP-Seq TF binding maps.

The data comparison is now shown in Supplementary Fig. 2d, e, which revealed that our data fits well with Goode's. We found their results might be very valuable for exploring essential TF collaborations for establishing the SCL^{high} state, and discussed this in the manuscript.

2) Scialdone et al (Nature 2016) reported genes associated with the initiation of gastrulation in mouse embryos. It will be important to see the degree of overlap with the Foxh1 KO data reported in this paper here, given the known role of Foxh1 in early embryo patterning.

T-related genes reported in Scialdone et al's work are highly enriched in downregulated genes in Δ Foxh1-pr cells compared to wild type "pr" cells. The results are shown in Fig. 4i and Supplemental Fig. 4e.

3) I could not access the GEO accession number in the paper. It will be important for this to be made public if the paper gets accepted. I would have liked to see how they submitted their CrispR screen counts. I realize that the paper contains supplementary tables but are these also attached to the GEO submission (e.g. are there some processed data for the CrispR sequence reads; it would be useful if there are).

Please use the following link to access to GSE85641:

<https://www.ncbi.nlm.nih.gov/geo/query/acc.cgi?token=afwlvukufzazvkb&acc=GSE85641>

4) It would be good to provide a little more discussion of the previously described interplay between Ets and Fox transcription factors (for example work by Sarah de Val). It seems to me that there could be feed forward loops going on, which could be a nice speculation.

We included this point in the discussion.

5) Line 94: hemangiogenic is misspelled

We corrected this mistake.

References

1. Choi, K., Kennedy, M., Kazarov, A., Papadimitriou, J.C. & Keller, G. A common precursor for hematopoietic and endothelial cells. *Development* **125**, 725-732 (1998).
2. Chung, Y.S., *et al.* Lineage analysis of the hemangioblast as defined by FLK1 and SCL expression. *Development* **129**, 5511-5520 (2002).
3. Russ, A.P., *et al.* Eomesodermin is required for mouse trophoblast development and mesoderm formation. *Nature* **404**, 95-99 (2000).
4. Lee, D., *et al.* ER71 acts downstream of BMP, Notch, and Wnt signaling in blood and vessel progenitor specification. *Cell Stem Cell* **2**, 497-507 (2008).

Reviewer #1:

Remarks to the Author:

The authors have addressed all my comments adequately.

Reviewer #2:

Remarks to the Author:

I am happy with the way the authors have addressed my questions.

REVIEWERS' COMMENTS:

Reviewer #1 (Remarks to the Author):

The authors have addressed all my comments adequately.

--

Reviewer #2 (Remarks to the Author):

I am happy with the way the authors have addressed my questions.

We thank the reviewers for accepting our revision work.